# Concordance of p16^INK4a^ and E6*I mRNA among HPV-DNA-Positive Oropharyngeal, Laryngeal, and Oral Cavity Carcinomas from the ICO International Study

**DOI:** 10.3390/cancers14153787

**Published:** 2022-08-04

**Authors:** Marisa Mena, Xin Wang, Sara Tous, Beatriz Quiros, Omar Clavero, Maria Alejo, Francisca Morey, Miren Taberna, Xavier Leon Vintro, Belén Lloveras Rubio, Llúcia Alos, Hisham Mehanna, Wim Quint, Michael Pawlita, Massimo Tommasino, Miguel Angel Pavón, Nubia Muñoz, Silvia De Sanjose, Francesc Xavier Bosch, Laia Alemany

**Affiliations:** 1Cancer Epidemiology Research Program, Catalan Institute of Oncology (ICO)-IDIBELL, L’Hospitalet de Llobregat, 08908 Barcelona, Spain; 2Centro de Investigación Biomédica en Red de Epidemiología y Salud Pública (CIBERESP), Instituto de Salud Carlos III, 28029 Madrid, Spain; 3Pathology Department, Hospital de Vic, 08500 Vic, Spain; 4Pathology Department, Hospital General de l’Hopitalet, L’Hospitalet de Llobregat, 08908 Barcelona, Spain; 5Oncology Department, Catalan Institute of Oncology (ICO)-IDIBELL, ONCOBELL, L’Hospitalet de Llobregat, 08908 Barcelona, Spain; 6Otorhinolaryngology Department, Hospital Sant Pau, 08026 Barcelona, Spain; 7Centro de Investigación Biomédica en Red de Bioingeniería, Biomateriales y Nanomedicina (CIBER-BBN), Instituto de Salud Carlos III, 28029 Madrid, Spain; 8Pathology Department, Hospital del Mar, 08003 Barcelona, Spain; 9Pathology Department, Hospital Clinic, 08036 Barcelona, Spain; 10Institute of Head and Neck Studies and Education, University of Birmingham, Birmingham B15 2TT, UK; 11DDL Diagnostic Laboratory, 2288 ER Rijswijk, The Netherlands; 12Division of Molecular Diagnostics of Oncogenic Infections, Research Program Infection, Inflammation and Cancer, German Cancer Research Center (DKFZ), 69120 Heidelberg, Germany; 13Infections and Cancer Biology Group, International Agency for Research on Cancer (IARC), 69372 Lyon, France; 14National Cancer Institute, Bogotá 111511, Colombia; 15ISGlobal, 08036 Barcelona, Spain; 16Universitat Oberta de Catalunya, 08035 Barcelona, Spain

**Keywords:** human papillomavirus, head and neck cancer, biomarkers

## Abstract

**Simple Summary:**

The utility of a diagnostic algorithm for the detection of HPV-driven oral cavity (OCC), oropharyngeal (OPC), and laryngeal (LC) carcinomas using HPV-DNA testing followed by p16^INK4a^ immunohistochemistry, taking E6*I mRNA detection as the reference standard, was assessed in HPV-DNA-positive formalin-fixed paraffin-embedded samples from 29 countries. The concordance of p16^INK4a^ and E6*I mRNA among 78, 257, and 51 HPV-DNA-positive OCC, OPC, and LC, respectively, was moderate to substantial in OCC and OPC but only fair in LC. A different p16^INK4a^ expression pattern was observed in those cases HPV-DNA-positive for types other than HPV16, as compared to HPV16-positive cases. We concluded that the diagnostic algorithm of HPV-DNA testing followed by p16^INK4a^ immunohistochemistry might be helpful in the diagnosis of HPV-driven OCC and OPC, but not LC. Our study provides new insights into the use HPV-DNA, p16^INK4a^, and HPV-E6*I mRNA for diagnosing an HPV-driven head and neck carcinoma.

**Abstract:**

**Background**: Tests or test algorithms for diagnosing HPV-driven oral cavity and laryngeal head and neck carcinomas (HNC) have not been yet validated, and the differences among oral cavity and laryngeal sites have not been comprehensively evaluated. We aimed to assess the utility of a diagnostic algorithm for the detection of HPV-driven oral cavity (OCC), oropharyngeal (OPC) and laryngeal (LC) carcinomas using HPV-DNA testing followed by p16^INK4a^ immunohistochemistry, taking E6*I mRNA detection as the reference standard. **Methods:** Formalin-fixed paraffin-embedded OCC, OPC, and LC carcinomas were collected from pathology archives in 29 countries. All samples were subjected to histopathological evaluation, DNA quality control, and HPV-DNA detection. All HPV-DNA-positive samples (including 78 OCC, 257 OPC, and 51 LC out of 3680 HNC with valid HPV-DNA results) were also tested for p16^INK4a^ immunohistochemistry and E6*I mRNA. Three different cutoffs of nuclear and cytoplasmic staining were evaluated for p16^INK4a^: (a) >25%, (b) >50%, and (c) ≥70%. The concordance of p16^INK4a^ and E6*I mRNA among HPV-DNA-positive OCC, OPC, and LC cases was assessed. **Results:** A total of 78 OCC, 257 OPC, and 51 LC were HPV-DNA-positive and further tested for p16^INK4a^ and E6*I mRNA. The percentage of concordance between p16^INK4a^ (cutoff ≥ 70%) and E6*I mRNA among HPV-DNA-positive OCC, OPC, and LC cases was 79.5% (95% CI 69.9–89.1%), 82.1% (95% CI 77.2–87.0%), and 56.9% (95% CI 42.3–71.4%), respectively. A p16^INK4a^ cutoff of >50% improved the concordance although the improvement was not statistically significant. For most anatomical locations and p16^INK4a^ cutoffs, the percentage of discordant cases was higher for HPV16- than HPV-non16-positive cases. **Conclusions:** The diagnostic algorithm of HPV-DNA testing followed by p16^INK4a^ immunohistochemistry might be helpful in the diagnosis of HPV-driven OCC and OPC, but not LC. A different p16^INK4a^ expression pattern was observed in those cases HPV-DNA-positive for types other than HPV16, as compared to HPV16-positive cases. Our study provides new insights into the use HPV-DNA, p16^INK4a^, and HPV-E6*I mRNA for diagnosing an HPV-driven HNC, including the optimal HPV test or p16^INK4a^ cutoffs to be used. More studies are warranted to clarify the role of p16^INK4a^ and HPV status in both OPC and non-OPC HNC.

## 1. Introduction

Apart from oropharyngeal carcinomas (OPC), oral cavity (OCC) and laryngeal carcinomas (LC) are the predominant subtypes of head and neck carcinomas (HNC), where a fraction of cases are driven by human papillomavirus (HPV) [1]. However, HPV attributable fractions (AFs) in non-oropharyngeal HNC (OCC—2.1%, LC—2.3%) are much lower than in OPC (30%) [1]. On the other hand, since the number of incident OCC and LC exceeds that of OPC [2], even low HPV-AFs for these sites translate to high absolute numbers of HPV-driven OCC or LC. Worldwide, it is estimated that approximately 52,000 new HNC cases are caused by a persistent HPV infection every year. Of these, 42,000 correspond to OPC, 5900 correspond to OCC, and 4100 correspond to LC [3]. Thus, around 20% of new HPV-related HNC cases are oral cavity and laryngeal tumors.

The prognostic advantage of HPV-driven OPC versus non-HPV-driven OPC is well established. On the other hand, the clinical implications of HPV-status in OCC and LC are not clear [4]. Thus, while it is widely accepted to test all newly diagnosed OPC for HPV tumor status in the clinical setting, routine testing of non-oropharyngeal HNC for HPV is not currently recommended [5].

The WHO campaign for eliminating cervical cancer [6] raises the possibility of elimination of other HPV-related cancers. Therefore, from a public health perspective, estimating HPV-AFs in non-oropharyngeal HNC is relevant to help assess the possible protective effect of HPV vaccination. Therefore, country- and type-specific baseline estimations of HPV-AFs in all HPV-related cancers, including non-oropharyngeal HNC, are warranted.

The identification of p16^INK4a^, a cell surrogate marker of HPV carcinogenic transformation, using immunohistochemistry is easy to implement and the most widely used standalone technique for HPV-driven OPC diagnosis in the clinical setting. A nuclear and cytoplasmatic 70% cutoff of stained cells is recommended [5].

However, to diagnose HPV-driven HNC at non-oropharyngeal sites, tests and test algorithms have not been validated so far, nor has there been an in-depth evaluation of the differences among non-oropharyngeal sites. While HPV E6/E7 mRNA detection is widely accepted as the reference standard test to elucidate the oncogenic role of HPV in the tumor, it is still challenging to implement in specific settings due to RNA fragmentation and degradation in paraffin-embedded tissue [7]. High-risk HPV RNA in situ hybridization is increasingly available as a clinical test on automated stainers and can detect highly fragmented RNA, but implementing the technique for HPV-driven HNC diagnosis is still under evaluation [7].

Moreover, despite the well-proven prognostic impact and clinical implications of HPV status in OPC and the wide use of p16^INK4a^ immunohistochemistry alone to assess HPV status in OPC, a fraction of p16^INK4a^-positive OPCs are HPV-DNA-negative, and HPV-DNA-negative/p16^INK4a^-positive OPCs do not show a prognostic advantage with respect to HPV-DNA/p16^INK4a^ double-negative cases [8,9]. These findings highlight the need for a test in addition to p16^INK4a^ immunohistochemistry to confirm HPV causality in OPC tumors, but no consensus for HPV testing has been reached yet. Moreover, most studies analyzing the discordance between p16 and HPV used an HPV-DNA-based test rather than the reference standard, an mRNA-based test.

HPV-DNA/p16^INK4a^ double testing is increasingly used for diagnosing HPV-driven OPC. In a meta-analysis of 11 studies [10], a sensitivity of 93% (95% CI 87–97%) and a specificity of 96% (95% CI 89–100%) for HPV-DNA/p16^INK4a^ double positivity were estimated. Yet, there is still limited information about the accuracy and prognostic value of this combination of biomarkers in OPC, and even less in non-oropharyngeal HNC.

The ICO international study estimated the fraction of cases attributable to HPV in 3680 formalin-fixed paraffin-embedded (FFPE) HNC tissues collected from pathology archives in 29 countries [1]. Estimates of the fraction of cases attributable to HPV based on positivity for HPV-DNA and for either HPV E6*I mRNA or p16^INK4a^ were 22.4%, 4.4%, and 3.5% for OPC, OCC, and LC, respectively [1,11].

Our aim in the present study was to assess the utility of a diagnostic algorithm for the detection of HPV-driven OCC, OPC, and LC of HPV-DNA testing followed by p16^INK4a^ immunohistochemistry, taking E6*I mRNA detection as the reference standard. For that, we evaluated the concordance of p16^INK4a^ and E6*I mRNA among HPV-DNA-positive OCC, OPC, and LC cases from the ICO international study.

## 2. Materials and Methods

### 2.1. Study Design

We conducted a statistical reanalysis of HNC cases positive for HPV-DNA, tested for p16^INK4a^ immunohistochemistry and HPV E6*I-mRNA detection in the ICO international study, a large cross-sectional international study coordinated by the Catalan Institute of Oncology (ICO) (Barcelona, Spain) in collaboration with DDL Diagnostic Laboratory (Rijswijk, The Netherlands) [1,11]. The Ethics Committee of the Catalan Institute of Oncology-ICO (Comitè Ètic d’Investigació Clínica de l’Hospital Universitari de Bellvitge, L’Hospitalet de Llobregat, Barcelona, Spain) formally approved the study on 9 September 2010 (protocol code PR101/08).

### 2.2. FFPE Block Processing and Histopathological Evaluation

Details of the protocol were described elsewhere [1]. Briefly, the block processing and histopathological evaluation of FFPE were performed as follows: all specimen processing was centralized in the ICO. The samples were sectioned, and at least four sections were obtained. The first and last sections were stained with hematoxylin/eosin (HE) and were used for histopathological evaluation. Intermediate sections were used to determine viral DNA, E6*I mRNA, and p16^INK4a^ detection.

### 2.3. HPV-DNA Detection and Genotyping

HPV determination was performed using PCR with the SPF-10 primer system followed by DEIA (DNA Enzyme Immuno Assay). DEIA-positive HPV-DNA samples were genotyped using the Line Probe Assay LiPA25 (Laboratory Biomedical Products, Rijswijk, The Netherlands); DNA quality was evaluated in all HPV-DNA-negative samples by testing for the tubulin-β gene. All DEIA and LiPA25 assays were performed at ICO.

### 2.4. HPV E6*I mRNA Detection

In all HPV-DNA-positive HNC, E6*I mRNA detection was performed by RT-PCR (real-time PCR). The E6*I mRNA assay targets 20 HPV types (HPVs 16/18/26/31/33/35/39/45/51/52/53/56/58/59/66/67/68/70/73/82). For each case, type-specific E6*I mRNA PCR was performed for all types detected by the SPF-10 PCR/DEIA/LiPA25 system that had at least one of the mRNA targeted types, in addition to HPV16. Quality control for mRNA detection was performed by detection of ubiquitin C mRNA. All E6*I mRNA assays were performed at DKFZ, Heidelberg, Germany.

### 2.5. p16^INK4a^ Immunohistochemistry

All HPV-DNA-positive cases were evaluated for p16^INK4a^ as previously described [1]. Three different nuclear and cytoplasmic staining cutoff values were considered: >25%, >50%, and ≥70%.

### 2.6. Statistical Analyses

Concordance percentage and kappa statistics were calculated to evaluate the agreement between p16^INK4a^ and E6*I mRNA among OCC, OPC and LC cases that were HPV-DNA-positive, HPV16-DNA-positive, and HPV-DNA-positive for types other than HPV16. Concordance percentage was estimated according to whether the condition np (1 − *p*) > 5 was met or not. If it was met, a normal distribution approximate method was applied [12]; if not, the estimates were obtained by the exact method based on binomial distribution [13]. The kappa statistic characterization was established as follows [14]: <0: poor; 0–0.20: slight; 0.21–0.40: fair; 0.41–0.60: moderate; 0.61–0.80: substantial; 0.81–1.0: almost perfect. The prevalence-adjusted bias-adjusted kappa (PABAK) statistic was explored when kappa was not valid (i.e., when the prevalence of a given response was very high or low, but low kappa values were observed in the 2 × 2 tables). The McNemar test *p*-value was also calculated to evaluate the distribution among the discordant cases. All statistical tests were two-sided, and statistical significance was set at a *p*-value of less than 0.05. All analyses were performed with STATA software, version 16.0 (Stata Corp, College Station, TX, USA).

## 3. Results

Figure 1 shows the disposition of cases included in the current analysis, in the context of the ICO international study. From the 3680 cases included in the main publication [1], 1264, 1090, and 1042 were OCC, OPC, and LC, respectively. Of those, a total of 78 (6.2%) OCC, 257 (23.6%) OPC, and 51 (4.9%) LC were HPV-DNA-positive and further tested for p16^INK4a^ and E6*I mRNA. Most HPV-DNA-positive cases came from Europe (237) and Central and South America (121, see Appendix A) and were males (71.5%) diagnosed between 2005 and 2009 (55.4%, see Appendix A). Mean age at diagnosis of HPV-DNA positive HNC cases was 58.5 years, being 60.2 for OCC, 58.5 for OPC, and 55.8 for LC. Conventional keratinizing squamous cell carcinoma was the most common histopathological diagnosis of HPV-DNA-positive OCC (55.1%) and LC (66.7%), whereas conventional non keratinizing squamous cell carcinoma was the most common histopathological diagnosis of HPV-DNA-positive OPC (42.4%).

The concordance between p16^INK4a^ at a cutoff ≥70% and E6*I mRNA was moderate among HPV-DNA-positive OCC (percentage of concordance 79.5% (95% CI 69.9–89.1%), kappa 0.59 (PABAK = 0.59)) and substantial among HPV-DNA-positive OPC (percentage of concordance 82.1% (95% CI 77.2–87.0%), kappa 0.36 (PABAK = 0.64)). Among LC cases, the concordance was only fair (percentage of concordance 56.9% (95% CI 42.3–71.4%), kappa 0.23 (PABAK = 0.14)) (Table 1). Lowering the p16^INK4a^ cutoff to >50% improved the concordance in all HN sites, although the improvement was not statistically significant (Table 1). The percentage of discordant cases (i.e., p16^INK4a^+/E6*I mRNA− or p16^INK4a^-/E6*I mRNA+) for HPV-DNA-positive LC was 42.1% when considering a p16^INK4a^ cutoff ≥70%, around two times higher than for HPV-DNA-positive OPC and OCC (17.9% and 20.5%, respectively). The same was also observed for p16^INK4a^ cutoffs of >25% and >50% (Table 1). For all anatomical locations and p16^INK4a^ cutoffs, the percentage of p16^INK4a^−/E6*I mRNA+ cases was higher than the percentage of p16^INK4a^+/E6*I mRNA− cases (Table 1).

HPV16 was the most common type among HPV-DNA-positive cases for all HN sites, although with lower proportions in OCC (69.0%) and LC (51.0%) than in OPC (83.0%, Figure 2). The next most common HPV type was HPV18 for OCC (4.3%) and LC (8.5%), and HPV33 (3.3%) for OPC. When considering as HPV-positive those cases double-positive for HPV-DNA/HPV-E6*I mRNA or HPV-DNA/p16^INK4a^, the percentage of HPV16-positive cases increased for OCC (96.0%, 85%) and OPC (89.0%, 87.0%) and decreased for LC (41.0%). These differences in HPV type distribution by HPV relatedness definition were statistically significant.

The concordance between p16^INK4a^ (all cutoffs) and E6*I mRNA decreased for all HN sites when only considering those cases that are HPV16-DNA-positive as compared to cases HPV-DNA-positive for any type, although this decrease was not statistically significant (Table 1). When only considering those cases HPV-DNA-positive for types other than HPV16, the concordance between p16^INK4a^ and E6*I mRNA was variable between HN sites and p16^INK4a^ cutoffs, albeit based on a very low number of cases. For most anatomical locations and p16^INK4a^ cutoffs, the percentage of discordant cases was higher for HPV16- than HPV-non16-positive cases. In particular, in OPC and OCC, p16^INK4a^−/E6*I mRNA+ cases dropped from 17.4% and 20.6%, respectively, among HPV16-positive cases to 6.1% and 0% among HPV-non16-positive cases, whereas the percentage of double-negative cases increased in both locations. This pattern was not observed in LC where an increase of p16^INK4a^+/E6*I mRNA+ and p16^INK4a^−/E6*I mRNA+ cases was observed (Table 1).

## 4. Discussion

Despite the imperative need to accurately assign HPV status in OPC patients given its clinical implications and, to a lesser extent, in non-oropharyngeal HNC, there is not yet consensus for HPV testing in HNC. We aimed to evaluate the utility of a two-step diagnostic algorithm which is already used in some clinical settings for detection of HPV-driven OPC. This algorithm consists of HPV-DNA testing followed by p16^INK4a^ immunohistochemistry in HPV-DNA-positive OPC, OCC, and LC cases, collated from the ICO international study [1]. For that, we considered E6*I mRNA detection as the reference standard and evaluated the concordance of p16^INK4a^ and E6*I mRNA among HPV-DNA-positive cases.

Although evidence on the need of adding a second HPV test to p16^INK4a^ immunohistochemistry to accurately assign HPV status in OPC is accumulating and HPV-DNA/p16^INK4a^ double testing is increasingly used for this purpose, there is still limited information about this combination of biomarkers [8,9]. Some specific aspects such as the optimal HPV test or p16^INK4a^ cutoffs to be used when using the combination of biomarkers have not yet been comprehensively evaluated.

We found that the concordance between E6*I mRNA and p16^INK4a^ among HPV-DNA-positive cases was moderate to substantial for OCC and OPC cases but only fair for LC. Albeit similar in magnitude, the concordance pattern was substantially different between OCC and OPC: For OPC, the concordant cases were mainly (75.1%) triple-positive (i.e., HPV-DNA/p16^INK4a^/E6*I mRNA-positive), whereas, for OCC, they decreased to 43.6%, and the percentage of HPV-DNA/p16^INK4a^/E6*I mRNA triple-negative cases increased from 7.0% in OPC to 35.9% in OCC, meaning that a much lower proportion of OCC than OPC cases are truly HPV-driven. The reduction in p16^INK4a^ and E6*I mRNA concordance in LC was mainly due to a decrease in the triple positivity. Noteworthily, the proportion of cases HPV-DNA0positive/p16^INK4a^0negative/E6*I mRNA0positive was much higher in LC than in OPC and OCC (39.2% vs. 16.0 % and 16.7%, respectively), strengthening the argument that p16^INK4a^ testing, even in combination with another HPV test, is not useful to diagnose an HPV-driven LC.

It has been previously reported that a higher proportion of both OCC and LC than OPC cases overexpress p16^INK4a^ but are HPV-negative [15,16]. In our study on HPV-DNA-positive HNC cases, p16^INK4a^ and E6*I mRNA discrepancies were substantially higher for LC than OCC. A possible higher misclassification of OPC cases as OCC than LC could partially account for those observed differences. Other aspects such as different type of tissue or different biology of the virus by anatomical site within the head and neck could also explain the observed differences [17]. The higher percentage of HPV-DNA-positive cases for types than other HPV16 in LC than in OPC and OCC could also explain our results, since a different p16^INK4a^ expression pattern was observed in those cases as compared to HPV16-positive or HPV-positive for any type.

Regarding studies comparing p16 performance for diagnosing an HPV-driven oral cavity, laryngeal, and oropharyngeal HNC, the group of Dr. Maura L. Gillison in the USA assessed the accuracy of two different criteria for p16 positivity in HPV-related OCC, OPC, and LC [18,19,20]. The two p16 criteria were (1) by the cutoff point of intense staining in >70% of the tumor, or (2) by H-score, derived from the cross product of the intensity of p16 staining (0–3) and percentage of tumor staining at a maximum intensity that was defined as positive if ≥60. The authors found that p16 immunohistochemistry as a standalone test, whether evaluated by the cutoff point of intense staining in >70% of the tumor or by H-score, had a high positive predictive value (PPV) for high-risk HPV E6/E7 mRNA in OPC [14]. In contrast, the assay had very poor PPV in LC and OCC [19,20]. A most recent study evaluated the characteristics and association of p16 and HPV in Thai patients with oropharyngeal and non-oropharyngeal HNC and found that discordance rates of HPV/p16 status were 23% and 7% for patients with oropharyngeal and non-oropharyngeal HNC, respectively [21].

We did not observe differences between p16^INK4a^ cutoffs for any site, although a non-statistically significant higher concordance between p16^INK4a^ and E6*I mRNA was observed for all HN sites when lowering the p16^INK4a^ cutoff from ≥70% to >50%. At the time the main study was published, the internationally agreed recommendations for testing all new OPC for high-risk HPV using a nuclear and cytoplasmatic 70% cutoff of stained cells [5] in the clinical setting were not yet published. A wide variety of definitions of a positive p16^INK4a^ immunohistochemistry test were used across studies [10]. Moreover, a 70% cutoff is not required in p16 testing in non-OPC HPV-related cancer sites [22]. A recent study examined and quantified p16 immunohistochemistry staining on cell blocks of cervical neck lymph nodes in OPC and used receiver operating characteristic (ROC) curve analysis to determine an optimal cutoff value with high sensitivity and specificity [23]. It found that a threshold of 15% p16 staining in cell block maximizes sensitivity and specificity. For the main study, a nuclear and cytoplasmatic 25% cutoff of stained cells was used [1], and differences in p16^INK4a^ accuracy by cutoff were not observed for any head and neck site. Thus, when considering HPV-DNA/p16^INK4a^ double test for diagnosis of HPV-driven OCC or LC, cutoffs other than 70% for p16 immunohistochemistry could be applied. Other authors have also pointed out that the 70% p16 cutoff used for HPV-OPC diagnosis may not be fully transferable to other HNC since the squamous epithelium is different within the HN area [24].

We found that, among HPV-DNA-positive HNC cases, the percentage of discordant cases was much higher for p16^INK4a^−/E6*I mRNA+ cases than for p16^INK4a^+/E6*I mRNA− cases. The observation was consistent for all HN sites and p16^INK4a^ cutoffs. Noteworthily, around 70%, 31%, and 52% of HPV-DNA+/p16^INK4a^− OPC, OCC, and LC cases, respectively, were E6*I mRNA+ and 2.5%, 8.1%, and 14.3% of HPV-DNA+/p16^INK4a^+ OPC, OCC, and LC cases, respectively, were E6*I mRNA−. These results may indicate the need for using and optimizing an mRNA-based test rather than a DNA-based test in combination with p16^INK4a^ for diagnosing an HPV-driven HNC, as also pointed out by other authors [9]. It would be important to evaluate if p16^INK4a^−/E6*I mRNA+ cases have different survival outcomes than p16^INK4a^+/E6*I mRNA− cases. A study from the US in OPC found that patients who were p16-negative but HPV mRNA-positive had a prognosis somewhat closer to double-positive patients, while those who were p16-positive, but HPV mRNA-negative had a prognosis closer to that of double-negative patients [9].

When considering only cases where HPV was the truly triggering carcinogenic agent (i.e., cases HPV-DNA/HPV-E6*I mRNA double-positive) or HPV-DNA/p16^INK4a^ double-positive cases, the percentage of HPV16-positive cases increased for OCC and OPC, but decreased for LC. Moreover, the percentage of discordant cases was higher for HPV16 than HPV-non16-positive cases. These results, if confirmed in other larger studies, may have implications when designing strategies of primary prevention of HPV-driven HNC with HPV vaccination. Moreover, current clinical practice algorithms for HPV testing make no effort to discern the impact of genotypes in HNC patients. A recent study found that HPV genotypes were unevenly distributed across anatomic sites of the head and neck, with an association of certain genotypes with small cell transformation [25].

The clinical significance of HPV status in OCC and LC is unclear, and previous studies have shown mixed results [4,26]. While some studies observed a prognostic advantage of HPV-positive cases in LC but not in OCC, others found the contrary or even worse survival outcomes for HPV-positive versus negative OCC [4]. However, most of the studies were not adequately powered to detect survival differences by HPV status, since HPV-AFs in LC and OCC were relatively small. Prospective, extensive studies assessing HPV status in LC and OCC are lacking. In addition to ours, research underway is currently delineating the role of HPV and p16 testing in non-oropharyngeal sites [24,27]. Noteworthily, Doll et al. [28] also considered three different p16 cutoffs (>25%, >50%, and ≥70%) when evaluating the prognostic significance of p16 in OCC.

In order to improve the diagnostic accuracy of the HPV-DNA/p16^INK4a^ double test, both in oropharyngeal and in non-oropharyngeal HNC sites, one could consider algorithms also involving tumor morphology [28], as already proposed for OPC [9]. However, it is important to keep in mind that HPV testing in non-oropharyngeal HNC is not currently recommended in the clinical setting.

Our study had several limitations. Not all cases were tested for p16^INK4a^ and E6*I mRNA, and the limited sample size did not allow assessing differences between anatomical subsites within the oral cavity, oropharynx, and larynx or between histological diagnoses. Of note, the current clinical guidelines on HPV testing in HNC do not recommend HPV testing in non-oropharyngeal HNC or in non-squamous OPC [5]. Lastly, there was a difficulty for the pathologists to classify the “gray zone”, i.e., cases with the percentage of cells overexpressing p16 between 25% and 70%.

In conclusion, the diagnostic algorithm of HPV-DNA testing followed by p16^INK4a^ immunohistochemistry might be helpful in the diagnosis of HPV-driven OCC and OPC, but not LC. A different p16^INK4a^ expression pattern in those cases HPV-DNA-positive for types other than HPV16, as compared to HPV16-positive cases, was observed. Our study provides new insights into the use HPV-DNA, p16^INK4a^, and HPV-E6*I mRNA for diagnosing an HPV-driven HNC including which are the optimal HPV test or p16^INK4a^ cutoffs to be used. More studies are warranted to clarify the role of p16^INK4a^ and HPV status in both OPC and non-OPC HNC.

## Figures and Tables

**Figure 1 cancers-14-03787-f001:**
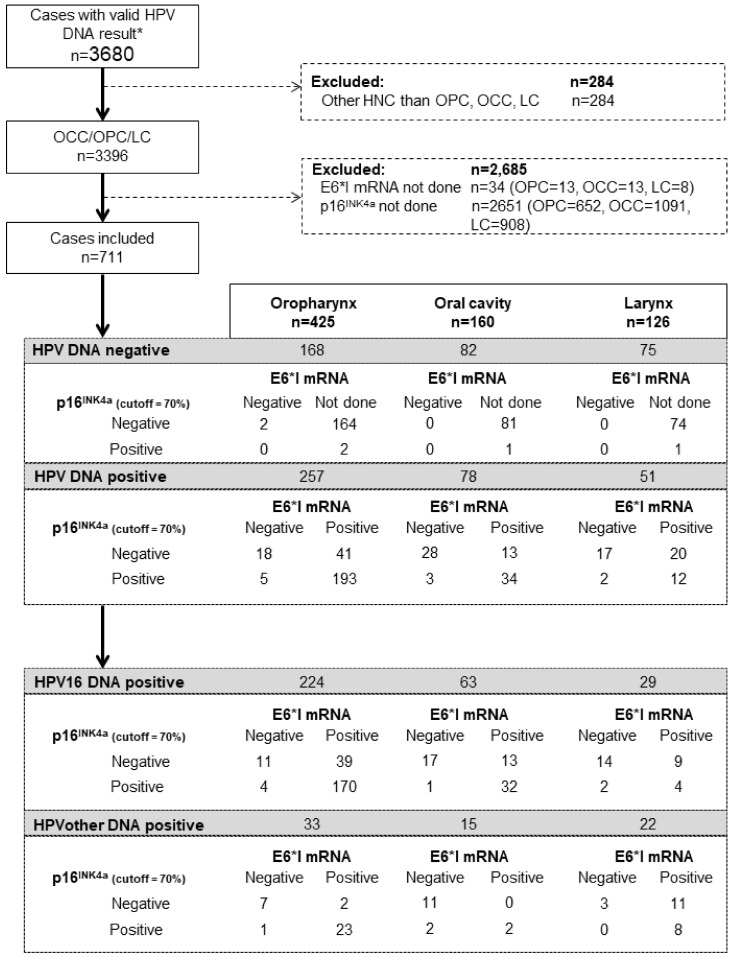
Flowchart of sample disposition and testing for HPV-related biomarkers. * Includes both cancers that were HPV-DNA positive and cancers that were HPV-DNA negative but tubulin positive.

**Figure 2 cancers-14-03787-f002:**
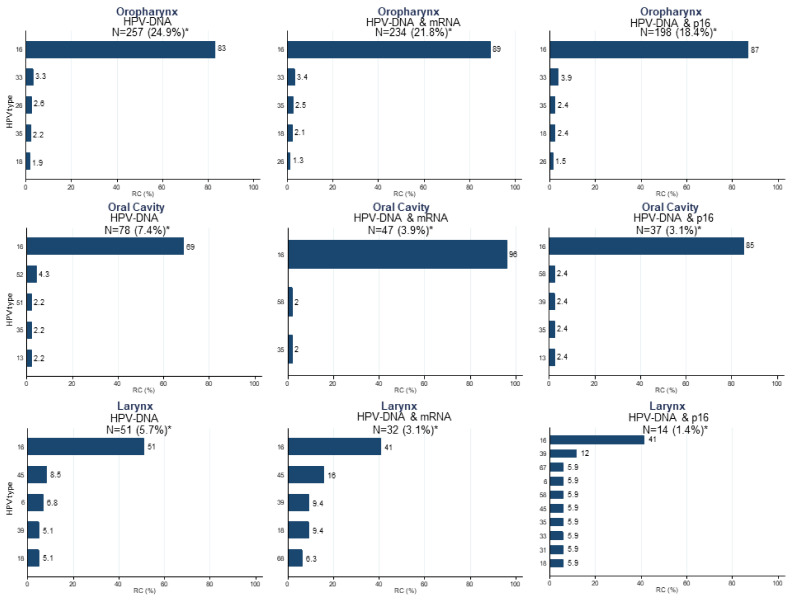
HPV type-specific relative contribution (RC) among HPV-positive cases when considering as HPV-positive HPV-DNA-, HPV-DNA/p16^INK4a^-, and HPV-DNA/E6*I mRNA-positive cases by anatomical location of the head and neck tumors. RC: relative contribution; * HPV-attributable fractions considering the oral cavity, oropharyngeal, and laryngeal cancer cases included in the main study [1], and a p16^INK4a^ cutoff of 70%.

**Table 1 cancers-14-03787-t001:** Concordance between p16^INK4a^ and E6*I mRNA among HPV-DNA-positive, HPV16-DNA-positive, and HPV-DNA-positive for types other than HPV16 by anatomical location of the head and neck tumors.

Head and Neck Carcinoma Cases	*N*	E6*I mRNA-Pos				E6*I mRNA-Neg				E6*I mRNA and p16^INK4a^ Concordance			
		p16^INK4a^-Pos		p16^INK4a^-Neg		p16^INK4a^-Pos		p16^INK4a^-Neg		% Concordance (95% CI) **	Kappa (95% CI)	PABAK	McNemar Test *p*-Value
		*n*	%	*n*	%	*n*	%	*n*	%				
**p16^INK4a^ Cutoff 70%**													
**OPC**													
HPV-DNA-pos	257	193	75.1	41	16.0	5	1.9	18	7.0	82.1 (77.2, 87.0)	**0.36 (0.22, 0.49)**	0.64	**<0.001**
HPV16-DNA-pos	224	170	75.9	39	17.4	4	1.8	11	4.9	80.8 (75.4, 86.2)	**0.26 (0.12, 0.41)**	0.62	**<0.001**
HPVother-DNA-pos *	33	23	69.7	2	6.1	1	3.0	7	21.2	90.9 (75.7, 98.1)	**0.76 (0.51, 1.00)**	0.82	1.000
**OCC**													
HPV-DNA-pos	78	34	43.6	13	16.7	3	3.8	28	35.9	79.5 (69.9, 89.1)	**0.59 (0.42, 0.77)**	0.59	**0.021**
HPV16-DNA-pos	63	32	50.8	13	20.6	1	1.6	17	27.0	77.8 (66.7, 88.8)	**0.55 (0.35, 0.74)**	0.56	**<0.001**
HPVother-DNA-pos *	15	2	13.3	0	0	2	13.3	11	73.3	86.7 (59.5, 98.3)	**0.59 (0.12, 1.00)**	0.73	0.500
**LC**													
HPV-DNA-pos	51	12	23.5	20	39.2	2	3.9	17	33.3	56.9 (42.3, 71.4)	**0.23 (0.03, 0.42**)	0.14	**<0.001**
HPV16-DNA-pos	29	4	13.8	9	31.0	2	6.9	14	48.3	62.1 (42.7, 81.5)	0.19 (−0.12, 0.51)	0.24	**<0.001**
HPVother-DNA-pos *	22	8	36.4	11	50.0	0	0.0	3	13.6	50.0 (26.8, 73.2)	0.17 (−0.03, 0.36)	0.00	**0.001**
**p16^INK4a^ Cutoff 50%**													
**OPC**													
HPV-DNA-pos	257	200	77.8	34	13.2	6	2.3	17	6.6	84.4 (79.8, 89.1)	**0.38 (0.24, 0.53)**	0.69	**<0.001**
HPV16-DNA-pos	224	177	79.0	32	14.3	5	2.2	10	4.5	83.5 (78.4, 88.6)	**0.28 (0.12, 0.44)**	0.67	**<0.001**
HPVother-DNA-pos *	33	23	69.7	2	6.1	1	3.0	7	21.2	90.9 (75.7, 98.1)	**0.76 (0.51, 1.00)**	0.82	1.000
**OCC**													
HPV-DNA-pos	78	37	47.4	10	12.8	4	5.1	27	34.6	82.1 (72.9, 91.2)	**0.64 (0.47, 0.81)**	0.64	0.180
HPV16-DNA-pos	63	35	55.6	10	15.9	2	3.2	16	25.4	81.0 (70.5, 91.4)	**0.59 (0.39, 0.79)**	0.62	**<0.001**
HPVother-DNA-pos *	15	2	13.3	0	0.0	2	13.3	11	73.3	86.7 (59.5, 98.3)	**0.59 (0.12, 1.00)**	0.73	0.500
**LC**													
HPV-DNA-pos	51	16	31.4	16	31.4	4	7.8	15	29.4	60.8 (46.4, 75.2)	**0.26 (0.02, 0.49)**	0.22	**0.012**
HPV16-DNA-pos	29	5	17.2	8	27.6	4	13.8	12	41.4	58.6 (39.0, 78.3)	0.14 (−0.21, 0.49)	0.17	**<0.001**
HPVother-DNA-pos *	22	11	50.0	8	36.4	0	0.0	3	13.6	63.6 (41.3, 86.0)	0.27 (−0.003, 0.55)	0.27	**0.008**
**p16^INK4a^ Cutoff 25%**													
**OPC**													
HPV-DNA-pos	257	200	77.8	34	13.2	6	2.3	17	6.6	84.4 (79.8, 89.1)	**0.38 (0.24, 0.53)**	0.69	**<0.001**
HPV16-DNA-pos	224	177	79.0	32	14.3	5	2.2	10	4.5	83.5 (78.4, 88.6)	**0.28 (0.12, 0.44)**	0.67	**<0.001**
HPVother-DNA-pos *	33	23	69.7	2	6.1	1	3.0	7	21.2	90.9 (75.7, 88.6)	**0.76 (0.51, 1.00)**	0.82	1.000
**OCC**													
HPV-DNA-pos	78	38	48.7	9	11.5	5	6.4	26	33.3	82.1 (72.9, 91,2)	**0.63 (0.46, 0.81)**	0.64	0.424
HPV16-DNA-pos	63	36	57.1	9	14.3	3	4.8	15	23.8	81.0 (70.5, 91.4)	**0.58 (0.37, 0.78)**	0.62	**<0.001**
HPVother-DNA-pos *	15	2	13.3	0	0.0	2	13.3	11	73.3	86.7 (59.5, 98.3)	**0.59 (0.12, 1.00)**	0.73	0.500
**LC**													
HPV-DNA-pos	51	18	35.3	14	27.5	5	9.8	14	27.5	62.7 (59.5, 98.3)	**0.27 (0.03, 0.52)**	0.25	0.064
HPV16-DNA-pos	29	5	15.6	8	25.0	8	25.0	11	34.4	55.2 (35.3, 75.0)	−0.04 (−0.38, 0.31)	0.00	**<0.001**
HPVother-DNA-pos *	22	13	59.1	6	27.3	0	0.0	3	13.6	72.7 (49.8, 89.3)	**0.37 (0.04, 0.71)**	0.45	**0.031**

OPC: oropharyngeal carcinoma; OCC: oral cavity carcinoma; LC: laryngeal carcinoma; *N*: number of cases; Pos: positive; Neg: negative. * Cases DNA positive for types other than HPV16. ** Can be estimated in two different ways, see Section 2. Statistically significant values are shown in bold.

## Data Availability

Data supporting reported results can be provided upon request to lalemany@iconcologia.net or mmena@iconcologia.net.

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
