# Peer review of "Concordance of p16INK4a and E6*I mRNA among HPV-DNA-Positive Oropharyngeal, Laryngeal, and Oral Cavity Carcinomas from the ICO International Study"

_cancers, 2022, doi:10.3390/cancers14153787_

Round 1
Reviewer 1 Report
The authors describe a diagnostic algorithm for the detection of HPV-driven OCC, OPC and LC using HPV DNA testing followd by p16INK4a immunohistocheemistry and taking E6*I mRNA as reference. They included 78 OCC, 257 OPC and 51LC in the evaluation and observed OCC and OPC HPV-driven diagnosis but not LC.
The introduction and methods section is clearly written and the results are adequately discussed.
Overall I can recommend this manuscript for publication.
Author Response
Thank you very much for your comments and positive review.
Reviewer 2 Report
The paper "Concordance of p16INK4a and E6*I mRNA among HPV‐DNA positive oropharyngeal, laryngeal and oral cavity carcinomas from the ICO international study" presents very relevant results from the ICO international study highlighting differences between oropharyngeal, oral cavity and laryngeal cancers regarding concordance of the analyses and the impact of different cut-off values for p16 staining. These differences are very relevant respective to interpretation of results form HPV DNA detection as well as p16 IHC.
However, there is an aspect I would like to see reflected. What about the positive rate when the data are structured the usual way according to algorithms proposed by many investigators, this means use of p16 IHC positivity as starting point for doing the analyses of HPV DNA and E6*I mRNA only in p16+ cases? It would be very nice to see these results surely demonstrating a high number miss-classified cases for instance within the supplement.
Author Response
Thank you very much for your review and relevant comment. Indeed, it would have been very interesting to present the positive rate when the data are structured the usual way according to algorithms proposed by many investigators, this means use of p16 IHC positivity as starting point for doing the analyses of HPV DNA and E6*I mRNA only in p16+ cases. However, we are afraid we are not able to present it as such, since our algorithm was starting the other way around and we only tested for p16 the HPV-DNA positive cases. However, the percentage of HPV-DNA+ cases that were also p16+ but E6*I mRNA negative has been included in the discussion, as follows:
“…2.5%, 8.1% and 14.3% of HPV-DNA+/p16INK4a+ OPC, OCC and LC cases, respectively, were E6*I mRNA-.“
Reviewer 3 Report
Mena, Wang et al. have written a straightforward analysis of data from the large multi-country ICO study regarding the use of p16INK4a and E6*I mRNA as diagnostic indicators. The data is convincing and addresses known issues regarding relying on a single diagnostic indicator for HPV+ HNC. In addition, they address the understudied area of HPV+ OCC and LC. I recommend this manuscript for acceptance after the minor revisions below.
Introduction Comments:
Line 91: “citoplasmatic” should be “cytoplasmic” (occurs throughout the text see Line 294 & Line 302 for example)
Materials and Methods Comments:
Line 148: “performed by specific RT-PCR” – Is there something about this RT-PCR that makes it more specific than conventional RT-PCR? Otherwise I suggest the authors remove the word specific.
Line 154: The way this line is currently written, it is unclear whether all RNA extraction and E6*I mRNA detection was performed in Germany or whether this occurred for only a subset of the samples. If all of the RNA extraction and E6*I mRNA detection was performed in a single location, perhaps this line should appear earlier in the methods.
Figure 1 Comments:
Perhaps adding a flow chart arrow between the HPV DNA positive box and the HPV16 DNA positive and HPV other DNA positive boxes could help illustrate that the data from the HPV DNA positive box is being further categorized in HPV16 DNA positive and HPV other DNA positive boxes.
Author Response
Thank you very much for your review and relevant comments.
Introduction Comments:
Line 91: “citoplasmatic” should be “cytoplasmic” (occurs throughout the text see Line 294 & Line 302 for example)
Thank you for noticing this. The typo has been corrected throughout the text.
Materials and Methods Comments:
Line 148: “performed by specific RT-PCR” – Is there something about this RT-PCR that makes it more specific than conventional RT-PCR? Otherwise I suggest the authors remove the word specific.
The word specific has been removed.
Line 154: The way this line is currently written, it is unclear whether all RNA extraction and E6*I mRNA detection was performed in Germany or whether this occurred for only a subset of the samples. If all of the RNA extraction and E6*I mRNA detection was performed in a single location, perhaps this line should appear earlier in the methods.
All E6*I mRNA assays were performed at DKFZ, Heidelberg, Germany. This has been clarified in the methods as follows:
“All E6*I mRNA assays were performed at DKFZ, Heidelberg, Germany.”
Figure 1 Comments:
Perhaps adding a flow chart arrow between the HPV DNA positive box and the HPV16 DNA positive and HPV other DNA positive boxes could help illustrate that the data from the HPV DNA positive box is being further categorized in HPV16 DNA positive and HPV other DNA positive boxes.
A flow chart arrow between the HPV DNA positive box and the HPV16 DNA positive and HPV other DNA positive boxes has been added in figure 1 as suggested.